# A Brief Introduction to Magnetoencephalography (MEG) and Its Clinical Applications

**DOI:** 10.3390/brainsci12060788

**Published:** 2022-06-15

**Authors:** Alfred Lenin Fred, Subbiahpillai Neelakantapillai Kumar, Ajay Kumar Haridhas, Sayantan Ghosh, Harishita Purushothaman Bhuvana, Wei Khang Jeremy Sim, Vijayaragavan Vimalan, Fredin Arun Sedly Givo, Veikko Jousmäki, Parasuraman Padmanabhan, Balázs Gulyás

**Affiliations:** 1Department of CSE, Mar Ephraem College of Engineering and Technology, Marthandam 629171, Tamil Nadu, India; leninfred@marephraem.edu.in (A.L.F.); fredin.givo@yahoo.in (F.A.S.G.); 2Department of EEE, Amal Jyothi College of Engineering, Kanjirappally 686518, Kerala, India; appu123kumar@gmail.com; 3Department of ECE, Mar Ephraem College of Engineering and Technology, Marthandam 629171, Tamil Nadu, India; ajay@marephraem.edu.in; 4Department of Integrative Biology, Vellore Institute of Technology, Vellore 632014, Tamil Nadu, India; sayantan7@gmail.com; 5Cognitive Neuroimaging Centre, Nanyang Technological University, Singapore 636921, Singapore; pbharishita@gmail.com (H.P.B.); simw0035@e.ntu.edu.sg (W.K.J.S.); vimalan.vijay@ntu.edu.sg (V.V.); veikko.jousmaki@aalto.fi (V.J.); 6Lee Kong Chian School of Medicine, Nanyang Technological University, Singapore 636921, Singapore; 7Aalto NeuroImaging, Department of Neuroscience and Biomedical Engineering, Aalto University, 12200 Espoo, Finland; 8Department of Clinical Neuroscience, Karolinska Institute, 17176 Stockholm, Sweden

**Keywords:** magnetoencephalography (MEG), clinical application, brain network, brain connectivity, neurological disorder, electrophysiology, diagnostic, therapeutic, computer-aided algorithms

## Abstract

Magnetoencephalography (MEG) plays a pivotal role in the diagnosis of brain disorders. In this review, we have investigated potential MEG applications for analysing brain disorders. The signal-to-noise ratio (SNR_MEG_ = 2.2 db, SNR_EEG_ < 1 db) and spatial resolution (SR_MEG_ = 2–3 mm, SR_EEG_ = 7–10 mm) is higher for MEG than EEG, thus MEG potentially facilitates accurate monitoring of cortical activity. We found that the direct electrophysiological MEG signals reflected the physiological status of neurological disorders and play a vital role in disease diagnosis. Single-channel connectivity, as well as brain network analysis, using MEG data acquired during resting state and a given task has been used for the diagnosis of neurological disorders such as epilepsy, Alzheimer’s, Parkinsonism, autism, and schizophrenia. The workflow of MEG and its potential applications in the diagnosis of disease and therapeutic planning are also discussed. We forecast that computer-aided algorithms will play a prominent role in the diagnosis and prediction of neurological diseases in the future. The outcome of this narrative review will aid researchers to utilise MEG in diagnostics.

## 1. Introduction

A systematic analysis for the Global Burden of Disease (GDB) study reveals that, as of Y2016, neurological disorders were the major cause of Disability Adjusted Life Years (DALYs) and are the second leading cause of deaths [1]. Neurological disorders are associated with structural and functional abnormalities and can be identified by several diagnostic imaging tools. Generally, the decrease in brain volume as a result of cerebral atrophy is one of the common characteristics associated with structural abnormalities of brain for many neurological disorders. As an example, patients diagnosed with Parkinson’s disease show a slight decrease in substantia nigra in comparison with normal control subjects [2]. However, the degeneration of neurons of substantia nigra is significantly higher in patients with chronic Parkinson’s disease in comparison to the early stage of the disease.

Other main neurological disorders manifest in specific ways. Memory loss is the vital symptom of Alzheimer’s disease, affects older adults, and is the common cause of dementia. Epilepsy is characterized by random seizures and complex seizures involving loss of consciousness. Schizophrenia is a mental disorder that affects the way a person thinks, acts, expresses emotions, perceives reality, and relates to others, and the symptoms appear early in men. Autism is a mental disorder that sways how a person perceives and socializes with others, causing problems in social interaction and communication.

Multiple neuroimaging approaches, viz., magnetic resonance imaging (MRI), functional magnetic resonance imaging (fMRI), positron emission tomography (PET), electroencephalography (EEG), and magnetoencephalography (MEG) are used as diagnostics tool in medical imaging. These neuroimaging tools are used to identify the structural and functional changes in the brain, and can therefore assist in early diagnosis and disease prognosis. MRI, which offers a high spatial resolution (on a scale of millimetres), is used extensively as a clinical neuroimaging tool to identify both the structural and functional changes associated with neurological disorders. The temporal resolution of MEG was found to be better than fMRI and PET, since their resolution is on the order of seconds. MEG picks the fields generated by intraneuronal currents and hence provides a direct index of neuronal activity and synaptic current [3]. The neuroimaging tools of MEG and EEG are generally coupled together since they both reflect the electrophysiological phenomena occurring in the brain. They do not only provide direct structural information about the brain but also create a direct environment for understanding functional aspects of the brain with a high temporal resolution (on a scale of milliseconds).

The human brain is a complex organ that consists of about 86 billion neurons and over (2.42 ± 0.29) × 10^14^ synapses that assist in communication between the neurons [4]. The top layer of the brain, the cerebral cortex, is about 2–4 mm thick. Neuronal activity within the cerebral cortex is associated with action potentials in axons, neurotransmitters synapses, and postsynaptic currents in post-synaptic dendrites of the pyramidal cells. Post-synaptic primary currents, elicited by neuronal activity, are the primary sources of MEG and EEG signals [5]. Thus, associated electromagnetic fields are direct ways to monitor and evaluate cortical processing in resting-state or challenge conditions [6].

Oscillatory brain activity, i.e., brain waves, are generated as a result of synchronized neuronal activity which could potentially be a biomarker for various physiological functions and behavioural states. The cortical rhythms can be categorized in several ways [7]. Brain waves are classified into five major types based on their frequencies: Delta (0.5–4 Hz), Theta (4 Hz), Alpha (8–12 Hz), Beta (12–35 Hz), and Gamma (>35 Hz) [8]. MEG is used to detect the brain waves generated across different regions of the brain, and its amplitude can be calculated by Power Spectral Density (PSD) analysis.

MEG is an ideal candidate for clinical applications in patients who cannot undergo a stringent clinical procedure before and during imaging. The first MEG signal was measured in 1968 at the University of Illinois by physicist David Cohen using the copper induction coil as detector, which resulted in MEG signals [9] with inadequate signal to noise ratio (SNR). Later, the Super Conducting Quantum Interference Device (SQUID), developed by Zimmerman, was used, which increased the SNR of the MEG signals and thereby paved the way to acquire the MEG signals without signal averaging [10,11]. In commercial MEG equipment, several sensors were placed as an array, like a helmet model, which enhanced the effective measurement and the spatial resolution of the MEG signals. With the technological advancements over the last decades, e.g., whole head coverage, sophisticated noise suppression algorithms, and zero boil-off helium systems, MEG has been evolved as a multichannel whole-head MEG system and finds its application in clinical imaging. Modern MEG systems are equipped, e.g., with 306 sensors (Figure 1). One of the current designs includes sensors comprising magnetometers and gradiometers within one sensor element. Such a design facilitates improved signal-to-noise for nearby cortical sources, suppression of ambient noise, and even suppression of nearby artifacts produced by, e.g., vagus nerve stimulators, cardiac pacemakers, and deep-brain stimulators.

MEG has two clinical applications, i.e., localization of epileptic foci and pre-operative evaluation for brain surgery candidates. MEG is also an important neuroimaging tool for the therapeutic planning of several mental disorders and abnormality analysis such as epilepsy [12,13,14,15,16], autism [17,18], schizophrenia [19], stroke [20], head trauma [21], and monitoring of drug administration [22]. The autism patients, when subjected to eye-gaze processing, have impaired activity in the gamma frequency band [23]. The authors concluded that the participants with severe ASD have higher activity between left temporo-parieto-occipital regions at 0–15 Hz frequency and higher activity between right temporo-parieto-occipital regions at 30 to 45 Hz in the low gamma frequency [24]. The 16 children with ASD are very sensitive to illegal speech sequences when undergoing 504.63 Hz MEG recording. The participants are age- and gender-matched controls [25]. The ASD patients have low social behaviour and communication due to a pattern of lower gamma band coherence in angular and middle temporal cortical regions within the default mode network [26]. The study with DTI connectivity of the hemisphere containing the epileptic focus in WM fibres of mTLE patients was confined with a connectivity-based laterality model affecting these brain regions [27]. The authors studied the importance of language mapping with MEG and the need for localization and lateralization with the changes in language networks and to identify the speech and social communication cortices in the brain [28].

## 2. Setting up the MEG Experiment

The general steps involved in setting up MEG experiments are discussed in this section. The subject is placed in a dedicated HPI chair after checking for any metal implants, and in total seven electrodes are placed on the subject’s face and neck (Figure 2). Two vertical Electro-oculography (EOG) electrodes are placed: one above and one below the left eye and two horizontally on the left and right temples, to record the eye blinks and saccades. These signals will be used later to the eye movement artefacts. Two ECG electrodes are placed—one on the right abdomen and one below the left clavicle. The ground electrode is to be placed on the bone of the clavicle. All the electrodes are immobilized with microporous tapes [5].

Unlike other imaging modalities, viz., MRI, X-ray, and PET, MEG does not give any direct structural details of the brain in the form of an image. Furthermore, acquisition in MEG is performed with respect to the SQUID sensors instead of the subject. Therefore, state-of-the-art MEG systems include a sub-system to determine the position of the head with respect to the MEG sensors. To accurately find the source of the neural activity, the size, orientation, and movement parameters of the head during the course of the acquisition must be known. This is achieved by using dipole-like coils, called Head Position Indicator (HPI) coils, and by the process of digitizing the input data. During subject preparation, the four HPI coils are attached to the subject’s scalp in a predetermined position. After attaching the electrodes to record the eye blinks and saccades, the process of digitizing is initiated using a stylus to generate a 3D head model in Cartesian coordinates. During the MEG signal acquisition, the HPI coils are activated to generate a magnetic field source, which is localized by the MEG sensors.

Since the sensors are fixed permanently and the prior knowledge about their orientations and positions is known with respect to each other, the relative position of the head can be calculated using the positions of the HPI coils that can be very accurately determined in the 3D space. The HPI coil used for co-registration is depicted in Figure 3.

Further, anatomical landmarks, viz., three bony fiducial points (Nasion, left, and right pre-auricular points) and additional points are collected around the subject’s head. Thus, the digitizing process will provide information about the subject’s head orientation, position, and shape. The head position in MEG experiments is either measured at the start/end or during the course of the experiment. Generally, a head motion of about 5 mm is acceptable in MEG experiments. General guidelines to be followed before starting a MEG acquisition are well documented in [32], and some of the key points are listed below.

(a)An empty room measurement is recommended for about 2 min before and after the actual experiment.(b)Simultaneous ECG and EOG acquisition is advisable since it is used for artefact rejections and corrections during pre-processing the data.(c)For experiments involving muscle movements, a recording of muscle activity is advisable.(d)Suitable subject choice is a must following the exclusion criteria.(e)Position the participant as close as to the sensors.(f)Once the subject is positioned in the MEG system, it is recommended to perform 2 min of resting state measurement to ensure proper functioning of all the accessories.(g)3D anatomical MRI is advisable if the protocol contains source localization process.

## 3. MEG Signal Processing and Source Localization

After the acquisition of MEG signals, the data is pre-processed. This step is important as it removes any extraneous signals from the data originating from anywhere other than the neuronal activity. These artefacts are sorted into following three categories:(a)External magnetic field interference caused by sources like electric lines, traffic, and elevators.(b)System-related issues caused by defective sensors(c)Physiological artefacts arising from the subject, viz., eye blinks, cardiac pulsing, subject motion, based on their origin [32]. Some of the common artefacts in MEG acquisition are shown in (Figure 4).

Two strategies are followed in general to remove the artefacts. The first being visual inspection and/or programmatic detection to identify and remove system related artefacts, and the second is through systematic signal processing steps, which are generally employed to remove external and physiological artefacts.

### 3.1. Signal Space Separation [SSS]

The bio-magnetic signals generated in the brain when recorded suffers severe interference from internal and external sources. To mitigate the interference from external sources, e.g., from infrastructure or electrical lines, the MEG experiment is usually performed in a Magnetically Shielded Room (MSR), which reduces the interference of several orders of magnitude. However, a method is still needed to reduce the signals to a minimal level. During the past four decades several methods were developed to address this problem. Some of the methods are gradiometric coil configurations [34], Signal space projection (SSP) [35], reference sensors [36] and Signal Space Separation (SSS) [37,38] methods. SSS is a spatial method, which transforms the MEG signals acquired from multiple channels, over 300, into its uncorrelated basic components called subspaces, one component from source outside the other from inside the MEG sensors. After the separation of the components the signals are extracted from the acquired MEG data based on the geometry of the sensor configuration. This method was developed based on Maxwell’s equations and assumes that all the sensors are about 4 cm away from the magnetic field sources. One of the advantages of this method is that it does not modulate or alter the original distribution of the MEG signals across the sensors. SSS was found to be more proficient in improving the quality of MEG data than classical methods with less user intervention. This method proves to be robust and provides a shielding factor of about 150 and 50 for sources at 1 and 0.5 m, respectively [37].

However, the SSS method does not remove the artefacts whose sources are near the sensors, e.g., pacemakers/stimulators located in proximity of the sensors, since the magnetic fields produced by these sources are spatially complex in nature and exceed the sensor noise in amplitude. Another method, which is a temporal extension of the SSS (tSSS), was proposed to remove such artefacts generated from the nearby sources. This method assumes there is a difference in the temporal pattern between the brain signal and the artefacts. To perform tSSS, a prior SSS operation must have been performed on the raw data. The principle behind this is explained briefly as follows. When performing SSS on the data, the spatial frequency included in the process had an incomplete description of the artefact magnetic field, and it leaks into both the internal and external part of the SSS reconstruction. This leakage forms the basis for tSSS as the temporal pattern in both the subspaces (inside and outside the source) is similar and the brain signal is only the part of the inside source and does not leak into external SSS basis. The temporal pattern relevant to the artefact sources are eliminated by projecting them in the time domain extracted from the internal signals reconstructed from SSS. The final data after SSS and tSSS are the same if there are no sources of artefacts nearby the sensors [39].

### 3.2. Software Tools Used in MEG Data Processing

Since the MEG signal amplitudes are of the order of femtotesla (fT), it is sensitive to several artifacts from different sources. The artifacts are usually removed through pre-processing steps using sophisticated software. The source of artifacts in MEG signals is power line interferences, signals from equipment, mechanical vibrations, and activities outside the brain [40]. The Signal Space Separation- and temporal Signal Space Separation-based algorithm by MEGIN Oy (Helsinki, Finland) are used for the pre-processing of MEG signals, and epoch-based techniques identify and rejects epochs containing eye blinks, muscular artifacts, and sensor jumps [41]. In [42], a detailed study on data acquisition and analysis of EEG/MEG analysis was performed.

In addition to vendor specific data analysis packages, several open source tools are available for MEG data analysis and visualization. Here we list a number of those pack-ages (Table 1). NUTMEG is a software tool for the analysis of EEG/MEG signals and is compatible with other MATLAB toolboxes [43]. The NIRS AnalyzIR Toolbox was developed primarily to process the NIRS, it offers limited support for MEG datasets, and its application may be extended for multimodal image processing [44]. The Field Trip is a MATLAB-based free toolbox for analysis of EEG/MEG and other bio signals. There is no other GUI and user should have the prerequisite knowledge about MATLAB functions for handling that dataset [45]. Brain Dynamics and Cognition Laboratory in Lyon developed ELAN toolbox for the analysis of EEG/MEG and local field potentials. The fast execution time was obtained by optimized algorithm and compiled C code. ELAN Annotation Format (EAF) are supported MATLAB functions [46]. SPM is also a MATLAB-based toolbox and can be integrated with fieldtrip toolbox for EEG/MEG analysis. The dynamic casual modelling technique combines neural modelling with data analysis for dealing with wide variants of responses [47]. EMEGs (ElectroMagnetoEncephalography software) is a MATLAB toolbox for pre-processing, analysis of, and visualization of electromagnetic data and can be used as a plug-in interface [48]. Brainstorm is an open-source software that emphasizes cortical source estimation and the analysis is done with the integration of MR images [49]. ERP WAVELAB was proposed for the multi-channel time-frequency analysis of EEG and MEG signals [50]. In [51], the characteristics of various software toolboxes available for the analysis of MEG and EEG datasets are presented.

## 4. Clinical Application

The attempt to identify potential clinical applications of MEG has been ongoing [53] since the early phases of the MEG. The first clinical applications were demonstrated for epilepsy patients [54,55]. In its infancy, MEG with one sensor was recorded simultaneously, with EEG serving as a trigger for averaging purposes. This attempt had exhibited the potential of MEG in identifying the sources of epileptiform spikes and its spread to the other hemisphere in about 20 ms. Several forms of epilepsy and multiple sources of epileptic activity in patients were identified. Further the co-registration of the evoked responses to 3D MRI resulted in preoperative planning [56]. Alzheimer’s disease is a chronic disorder that profligates the brain cells in [57], 3D convolutional neural network was proposed for the diagnosis Alzheimer’s from 2D MR data, and the architecture were termed as Alzheimer’s network. Schizophrenia alters the composition of grey matter; there are challenges in the detection of grey matter in brain from volumetric MR data. A deep learning model was proposed in [58] for the identification of Schizophrenia from structural MRI data. Vocal cord disorder and speech defacement are the early symptoms of Parkinson disease, and various machine-learning techniques are employed in [59] for the detection of Parkinson disease. In this section, the application of MEG in various neurodegenerative diseases will be discussed.

### 4.1. Epilepsy

According to WHO, around 50 million people across the globe are affected by epilepsy—a chronic non-communicable brain disease. The symptoms include recurrent seizures and involve either the whole brain (generalized) or part of the brain (focal), accompanied by loss of consciousness and control of bowel or bladder function in certain cases.

The functional neural network topology for epilepsy subjects is different from the healthy subjects, especially in the theta band. In epilepsy, some regions of the brain generate abnormal electrical signals, which in turn create magnetic signals and therefore can be detected by MEG. Scott B. Wilson et al. performed a detailed study on the detection of a spike in neuro signals [60]. The spatial accuracy of MEG is good since distortion is less in MEG signals when compared with the EEG signals. The EEG and MEG epileptic spikes are identified in the time and frequency domain methods [61]. MEG finds its application in the detection of interictal epileptiform discharges and localizing functional cortices, to guide neurosurgical procedures [62]. The linear discriminant analysis (LDA) classifier was used for the classification of MEG data obtained from 15 healthy subjects and 18 epilepsy patients [63]. A two-stage algorithm comprising beamforming by virtual sensors and time-frequency analysis by Stockwell transform was used to detect the high-frequency signals that help in the presurgical planning [64]. The structural images from imaging modalities, viz., MRI, PET, and SPECT data of the patients, are usually co-registered with MEG for the epilepsy surgery evaluation [65]. The genetic algorithm with K- nearest neighbor was used for the classification of epileptical MEG spikes [66]. In [67], coherence analysis for epilepsy patients was performed on MEG data. The nonlinear signal analysis was found to be effective in the analysis of Idiopathic Generalized Epilepsy (IGE) and from healthy volunteers of 10 subjects [68]. MEG, being a non-invasive technique, is a potential tool for epilepsy surgery evaluation and can determine the abnormalities observed in structural and functional mapping [69]. MEG is an effective tool for children with intractable focal epilepsy to determine the surgical candidacy and focal cortical resection to stop seizures [70]. Recently [71] an optically pumped MEG (OPM) has been used to study epilepsy. The OPM, a cryogen-free MEG system, can be directly placed over the scalp and is invariant to head motion. The performance of OPM was found to be similar to EEG in the detecting the markers of epilepsy. The OPM, on the other hand, utilizes only 20–50 sensors placed over the suspected region; therefore, this method does not provide whole head coverage. (Figure 5)

### 4.2. Alzheimer’s Disease (AD)

AD is one of the common neurodegenerative diseases that affects over 60–70% of the 47.5 million people with dementia across the globe, according to Dementia fact sheet WHO. The onset of the disease happens several years before it is clinically diagnosed. AD is characterized by three stages. The first stage is a pre-clinical phase, which lasts for over a decade. During this stage, an abnormal biomarker pattern is exhibited and low amyloid $\beta_{42}$ in cerebrospinal fluid (CSF) or increased tracer detection is shown in PET imaging. Towards the end of the first stage, neurodegeneration or injury is found. The second phase exhibits Mild Cognitive Impairment (MCI) and the third state is Alzheimer’s dementia. Therefore, there is need for the early intervention during the pre-dementia phases.

The identification of biomarkers for the neurodegenerative diseases will help in early diagnosis or the onset of the disease. For AD, Amyloid-β deposition is a well-established clinical biomarker, which starts several decades before the onset of the AD [73,74,75]. However, several research studies are being conducted in search of electrophysiological/electromagnetic markers for AD, which could assist in evaluating the early diagnosis of the pre-dementia phases. The report by [76] had identified a relation between the Amyloid-β deposition and the changes in the regional brain wave patterns, using resting state MEG as a technique. Briefly, the findings are: (a) there is an increase in alpha band activity in the medial frontal area, which reflects the Amyloid-β deposition (b) there is an increase in delta band power in the medial frontal area, showing that there is a regional decrease in glucose metabolism and showing a symptom of disease progression within the AD phases, (c) a global decrease in theta band activity only exhibits a general cognitive decline, not specific to AD. Thus, these findings are promising in that MEG could be a potential tool to provide electrophysiological biomarkers for the determination of predementia phases of AD.

In [77], the spectral property of MEG signals was utilized to distinguish between control, MCI, and Alzheimer’s disease (AD) subjects. In this study, a mean frequency approach was adapted before which the power spectral density of the MEG signals was calculated by the Fourier Transform of the autocorrelation function. The mean frequency is shown to decrease significantly in MCI patients and the values are intermediate between controls and AD patients. Another approach based on the MEG background activity was performed in [78]. In this study nonlinear techniques based on sample entropy (SampEn) and Lempel Ziv (LZC) complexity were used for the analysis of AD and control subjects using a 148-channel whole-head MEG system. The results suggest that for the AD subjects the MEG background activity revealed an increase in regularity and decrease in complexity, demonstrating that the neuronal dysfunction in AD can be identified by MEG background activity. Another study based on a missing stimulus paradigm was conducted in [79]. In this study, the subjects were exposed to short beep tones at certain intervals, and the tones were omitted randomly in the 160-channel MEG system. One of the advantages of the missing stimulus paradigm is the subjects need not pay attention to the stimuli. It was found that the amplitude of the average waveform is lower for AD subjects when compared with the control group. The study also concludes that the absence of the response to the omitted tone event could be an index for the early diagnosis of AD.

The MEG provides a 3D mapping of the brain so that functional connectivity of regions of the brain can be analyzed for the diagnosis of disorders [80]. The spectral coherence and cross mutual information function (CMIF) properties of the MEG waveform were used for the brain connectivity analysis in AD subjects [81]. The spectral entropy and statistical complexity measures were used for the analysis of MCI and AD subjects. The MCI subjects depict the intermediate pattern of abnormalities between control and AD subjects [82]. The MEG delta mapping was used for the analysis of 35 AD patients, 23 MCI patients, and 24 healthy control patients [83]. The Bayesian factor analysis algorithm was used for the analysis of MEG signals, using the Hadoop ecosystem [84]. The multilayer neural network was used for the classification of AD subjects with classification accuracy (78.39%) and sensitivity (91.11%) [85]. The MEG was found to be effective in the differential diagnosis of AD and major depression-related cognitive decline in the elderly subjects [86]. (Figure 6)

### 4.3. Schizophrenia

Schizophrenia is a severe mental disorder that has affected over 20 million people worldwide. A person affected by schizophrenia often exhibits the following symptoms: distortions in thinking, emotions, perception, language, behavior, sense of self, delusions, and hallucinations. Several research studies have been conducted to identify the region of the brain that is related to the symptoms of Schizophrenia, but the neural mechanism for the disease is yet to be identified. MEG can act a potential tool in identifying the electrophysiological marker for Schizophrenia. The disturbances in the oscillatory wave patterns can provide some insight regarding the symptoms or onset of the disease. Resting state MEG had been used to study schizophrenia and the findings suggests that the pathophysiology of schizophrenia can be correlated to the neural abnormalities in synchronized oscillatory activity [87]. This is complemented by EEG studies that the increase in delta, theta, and beta waves and the decrease in alpha power patterns had been identified in Schizophrenic patients [88]. However, there is poor reproducibility because of the sample characteristics, techniques adapted, and spatial distribution across the studies conducted [89,90]. Maor Zeev-Wolf et. al. [91] had employed the resting-state MEG to study the wave patterns in control and schizophrenic patients. Their findings suggest that high alpha power was negatively correlated with positive symptoms and beta power was positively correlated with the negative symptoms (Figure 7). The study concludes that different neural mechanisms may underplay in positive and negative symptomatic patients.

The MEG signal pattern was able to discriminate the schizophrenia patients from healthy subjects; 248-channel MEG signal analysis was performed on six healthy and six abnormal cases [92]. The gamma band activity can significantly differentiate healthy and schizophrenic patients, and a MEG measurement was taken on 15 schizophrenia and 15 healthy subjects. The recording was made while performing a complex mental arithmetic task and at rest. The gamma power was observed as high in healthy cases when subjected to the mental arithmetic task, while in the case of schizophrenia patients, less gamma power was observed regardless of the task. In [93], two techniques were used to analyze healthy and schizophrenia cases, based on the estimation of a number of dipoles in the delta and theta frequency and distribution, sources of slow wave activity. The beta and theta band activity were low in schizophrenia subjects and a higher number of slow wave generators was observed in certain areas of the brain. The dipole density plot was determined for healthy and schizophrenia cases, and for the diseased subjects there was an increase in the absolute dipole values in both the hemispheres [94]. The mismatch negativity is defined as the brain response subjected to deviations within a sequence of repetitive auditory stimuli and it was found to be absent in patients with schizophrenia. The fMRI and MEG data were combined for the analysis of healthy and affected subjects [95]. The diminishing alpha waves is a symptom of schizophrenia based on the analysis of 10 patients with schizophrenia and 18 healthy subjects [96]. The neuronal dynamics plays a vital role in the analysis of schizophrenia and neural synchrony was impaired in the schizophrenia subjects [97].

The MEG recordings of the gamma band activity of schizophrenia patients reveal the overactivity in the right frontal and right frontotemporal regions under cognitive demands (45 ± 71 Hz) (Figure 8. The gamma band activity was poor in frontotemporal, posteriotemporal, and occipital sites for the 60 to 71 Hz irrespective of the task [92]. The schizophrenic patients exhibited more enhanced activity in the low frequency bands (within the delta and theta frequency ranges) than the control subjects. Using the dipole density plot (DDP) method, the dipole localization was determined, and the results were superimposed on the MR images as isocontour lines [94]. The absolute dipole values measured in both hemispheres in schizophrenic patients were found to be high. The MEG and fMRI were coupled to study the altered neural responses to basic sound processing at the level of planum temporale in a group of schizophrenic patients and to correlate with the morphological changes in this region [94]. The MEG recording was done on schizophrenic patients during an auditory oddball task to investigate alpha brain activity related to selective attention to target stimuli and selective inhibition of irrelevant stimuli. The MEG-coherence source imaging (CSI) technique was employed to study and compare the brain oscillations (biomarkers) in normal subjects, and the schizophrenia patients were found to have an increased region of coherence [98]. The EEG and MEG focus on five measures: P50 auditory sensory gating, pre-pulse inhibition of the startle response (PPI), mismatch negativity (MMN), auditory P300, and gamma band oscillations. The measure indicates neuro defects such as inhibitory failure, aberrant salience detection, and impaired neural synchrony, which support the presence of higher-order cognition [99]. The static and dynamic connectivity measurements have been made by MEG-fMRI and the combined features have been used for the classification of schizophrenia subjects [100]. A pico-Tesla (pT) (1pT-10-12T)-TMS electronic device was developed to increase the (2–7 Hz) abnormal frequencies of the recorded MEG for patients with migraine, depression, or schizophrenia towards frequencies of less or equal to its frequencies of the alpha frequency range (8–13 Hz) [101]. The resting-state MEG can distinguish the different types of schizophrenia. The significant dysfunction in resting state connectivity is correlated with cognitive dysfunction and may cause differences in behavior and clinical presentation between subtypes of schizophrenia [102]. Although the alpha-band and baseline spectrum remain intact, gamma-band power at sensor level in schizophrenia patients during stimulus processing was found to be reduced. In schizophrenia subjects, high-frequency oscillations during visual processing were identified [103].

### 4.4. Parkinson Disease (PD)

As of 2016, 6.1 million people were affected by Parkinson’s disease globally, and it is the second most common neurodegenerative disease after AD. The neuropathological marker of PD is the deposition of Lewy bodies, especially alpha synuclein. PD affects the nigrostriatal dopaminergic neurons, causing them to be lost, resulting in dysfunctions of motor activities. As the disease progresses, it spreads from brainstem to other cortical regions in the later stages [104]. Therefore, PD is a whole brain disease, causing functional disturbances both in cortical and subcortical brain regions. Clinically, PD is characterized by its motor and non-motor symptoms. MEG is a potential tool for the diagnosis of the symptoms exhibited by PD in a non-invasive fashion. Since MEG offers a high temporal resolution, it can be employed to study the neural activity and the functional connectivity of the whole brain in patients with PD [105].

MEG was able to detect the thalamocortical dysrhythmia that is responsible for neurogenic pain, tinnitus, Parkinson’s disease, or depression [106], under resting state conditions. The subjects show low frequency theta due to resonant interaction between thalamus and cortex, which happens as a result of the low-threshold calcium spike bursts by thalamic cells due to hyperpolarization. In another study, MEG was acquired for PD patients and the results exhibited abnormal rhythmic activity, as shown by low frequency and high amplitudes. External magnetic stimulation (EMS) was performed with 1–7.5 pT and frequency in resonance with the alpha band (8–13 Hz) on the left-right temporal, frontal-occipital, and vertex for about 2 min. In PD patients a faster attenuation of the MEG activity was absorbed. The study concludes that the neural dynamics is strongly influenced by EMS.

Another study based on resting state MEG and power spectral density analysis was conducted in [107], and the results showed that the theta, beta, and gamma bands were characterized by slow resting-state potential in demented, non-demented PD, and healthy elderly controls. The study finds that in the non-demented patients the theta power was increased and beta power was decreased in comparison to the control subjects. However, in demented PD subjects, an increase in delta and a decrease in alpha and beta power was observed. Further, the study concluded that PD can be characterized by the slowing of resting-state brain activity in delta and alpha bands.

Further, MEG plays a crucial role in the analysis of biological neural-network functionality in the case of neurodegenerative disorders that have exhibited abnormal oscillatory and disturbed neural activity [108]. Also, MEG study reveals that the increased resting-state cortico-cortical functional connectivity in the 8–10 Hz alpha range is a feature of PD [109].

The inferences from Figure 9 are as follows: red color area indicates the statistically significant increase of relative power in the eyes open condition, blue color area indicates the significant decrease of relative power in the eyes open condition. The color intensity depicts the magnitude of change (light = 0–25%; middle = 25–50%; dark = >50%). A decrease in power is observed in the control group and was absent in PD patients.

### 4.5. Preoperative Evaluation

The above-mentioned applications of MEG are all largely for diagnostic and classification purposes. Additionally, the usage of MEG as a preoperative evaluation will become increasingly prominent and necessary. A common scenario will be the need for the sensorimotor mapping of the primary motor and sensory cortices (SM1) of brain tumor patients. Accurate localization of any function of interest to cortex enables optimal neurosurgical tumor resection and minimizes post-operative deficits. Currently, fMRI is largely the preferred choice of functional mapping in such cases. However, there are several reasons why MEG should be used instead, or in some cases, in conjunction with fMRI for preoperative evaluation. The first is that MEG is a direct physiological measurement of neural activity, which can tease apart brain–body interactions with more detail, as opposed to the blood–oxygen level dependency (BOLD) signal. MEG can distinguish between the two main types of brain–body interactions during motor movements: “cortex-kinematic” interaction, also known as corticokinematic coupling (CKC) and “cortex-muscle” interactions, also known as corticomuscular coupling (CMC). CKC is a signature of afferent signals, whereas CMC is mainly by efferent signals [98]. In one recording, MEG can acquire multiple neurophysiological processes such as evoked and induced magnetic responses, cortico-cortical coupling, and peripheral-cortical signals, which serve as several functional localizers [99]. This information will provide additional details to clinicians for a more thorough pre-operative assessment. The second reason is that fMRI measurements of atypical sensorimotor maps in brain-lesioned patients are more difficult to interpret than those of healthy subjects [100,101]. In such scenarios, it is recommended that MEG should be mandatory for precise source localization.

## 5. Inferences from the Narrative Study and Future Scope of MEG

The following are the inferences from the research studies on MEG. The MEG signals of interest are extremely small, several orders of magnitude smaller than other signals in a typical environment that can obscure the signal. Thus, specialized shielding is required to eliminate the magnetic interference found in a typical urban clinical environment. Patients need to remain relatively still during a MEG exam. [110]. Some of the MEG studies rely on sensor space data and some on source space data, hence a generalized framework is required in the analysis [111]. There is also a stimulus variation in most of the studies and a variation in output is observed prior to and after registration. MEG systems are less common when compared with the EEG and MRI systems, and standardization is also required in the quantitative analysis of MEG signals such as protocols, data collection, and data analysis. Few MEG public databases are available for research purposes when compared with the EEG and MRI databases. Table 2 shows the findings and Clinical Considerations of the MEG.

The usage of optically pumped magnetometers (OPMs) in the future will generate proficient results, however, the movement of OPM relative to the scalp during acquisition and recording will generate artifacts. The MEG system based on OPM can generate results with higher spatial resolution than the brain since multichannel recording is possible with potential measurement near the brain. In the future, the wearable MEG will make the system simple, and squid sensors are not required. The OPM-based MEG will be beneficial for acquiring the signals from children within a duration of 10 min, within the tolerance limits. Though MEG was proficient in the detection of neuro disorders, a specific pipeline for MEG data acquisition, processing, and analysis in a clinical setting is required, since it is not possible to compare the results of MEG studies. The utilization of optical co-registration in future will also improve the accuracy in the localization of potential.

## 6. Conclusions

This review provides a basic introduction to MEG and its clinical applications for neurodegenerative and associated diseases such as epilepsy, Alzheimer’s, schizophrenia, and Parkinson’s diseases. MEG was found to be an efficient tool for the diagnosis of brain disorders and for treatment planning. Additionally, MEG is a suitable clinical method in presurgical functional sensorimotor mapping and offers several advantages over fMRI. Its relevance in pre-operative evaluation is rising and will likely play a more crucial role in clinical settings in the near future. MEG is patient-friendly and new methods such as OPM decrease the complexity in pre-processing steps and enable efficient analysis of the signals. In some cases, early biomarkers in terms of power spectral density and background activity are demonstrated in the pre-dementia phases in AD. Together with other imaging modalities, viz., EEG and MRI, MEG acts as a potential tool for precise diagnosis and source localization.

## Figures and Tables

**Figure 1 brainsci-12-00788-f001:**
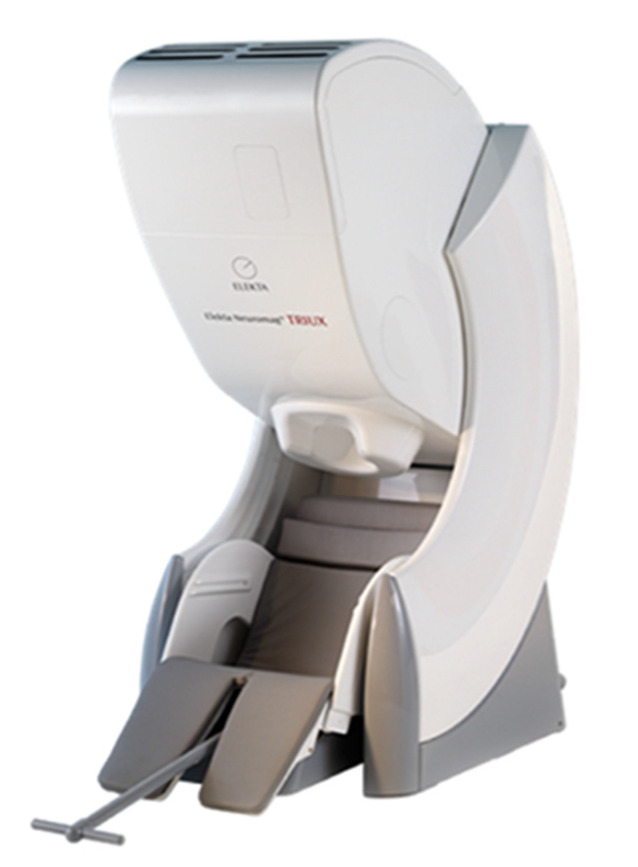
MEGIN Elekta Neuromag TRIUX MEG system with 306 SQUID sensors with an integrated 128 channel EEG. A state-of-the-art system with high tolerance for magnetic interference, improved subject comfort and zero Helium boil off. Reprinted from [29] (Copyright 2011 Elekta Oy).

**Figure 2 brainsci-12-00788-f002:**
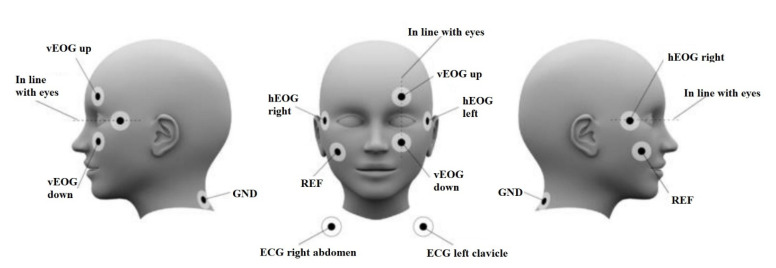
Placement of EOG and ECG for MEG Experiment. Reprinted from [30]. (Copyright 2017 NatMEG). Image reproduced with permission from *NatMEG*.

**Figure 3 brainsci-12-00788-f003:**
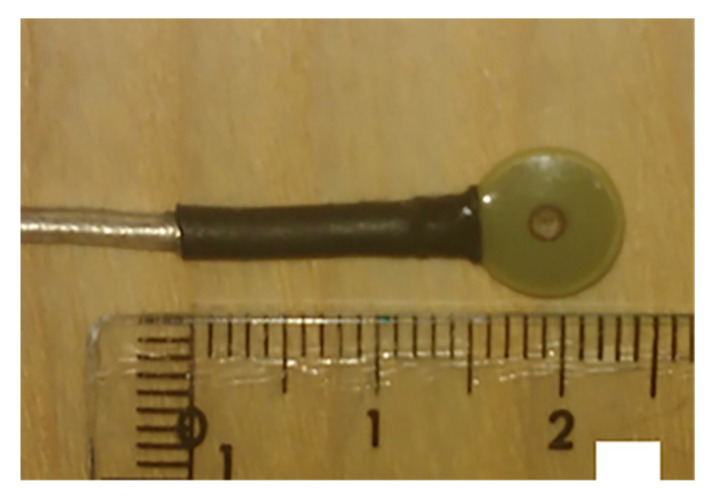
Head Position Indicator (HPI) coil used for co-registration purposes. Reprinted from [31]. (Copyright 2018 PLOS) Image reproduced as per terms of CC BY 4.0 license.

**Figure 4 brainsci-12-00788-f004:**
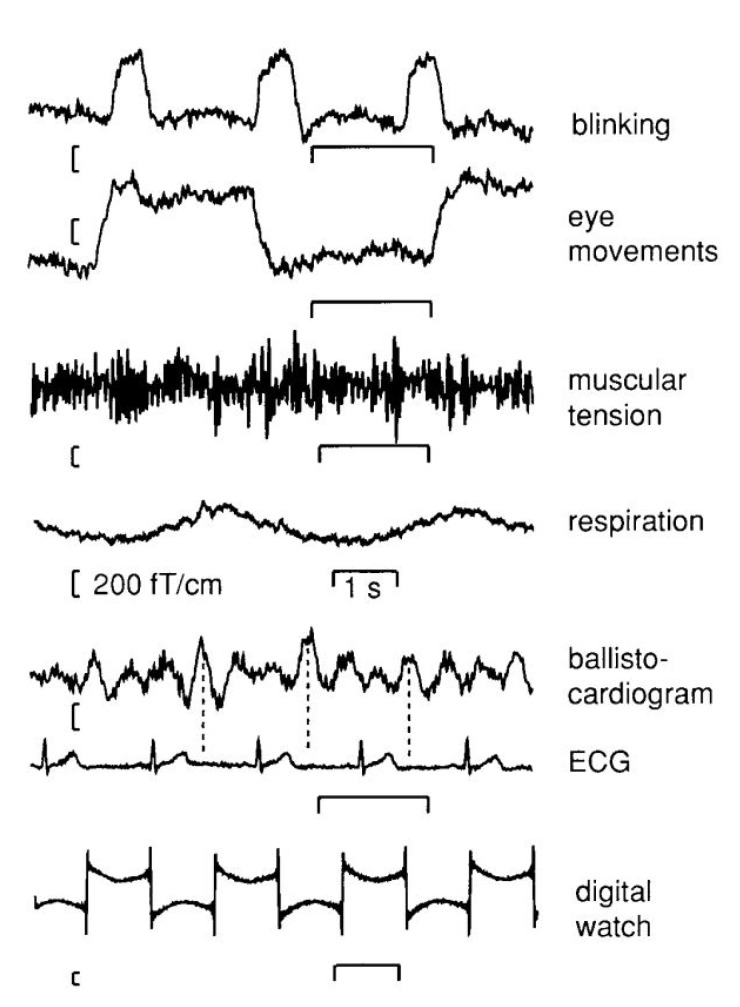
General artifacts encountered in MEG acquisition. Reprinted from [33]. (Copyright 2017 Oxford University Press) Image reproduced as per terms of CC BY 4.0 license.

**Figure 5 brainsci-12-00788-f005:**
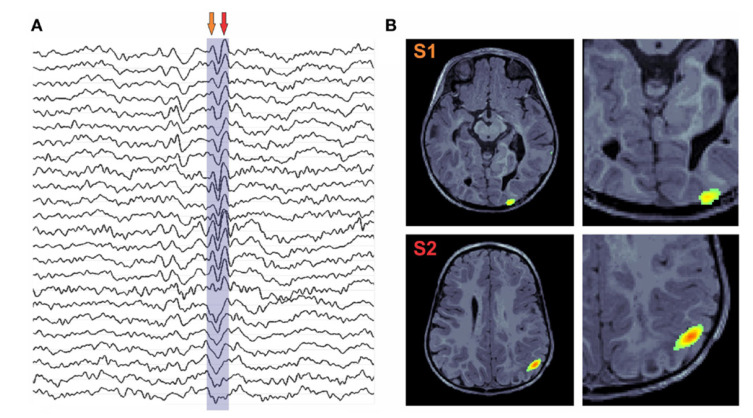
Shown here (**A**) are MEG traces of an epilepsy patient with spikes. In (**B**), Structural MRI with overlaid MEG activity. Reprinted from [72]. (Copyright 2014 Frontiers) Image reproduced as per terms of CC BY 3.0 license.

**Figure 6 brainsci-12-00788-f006:**
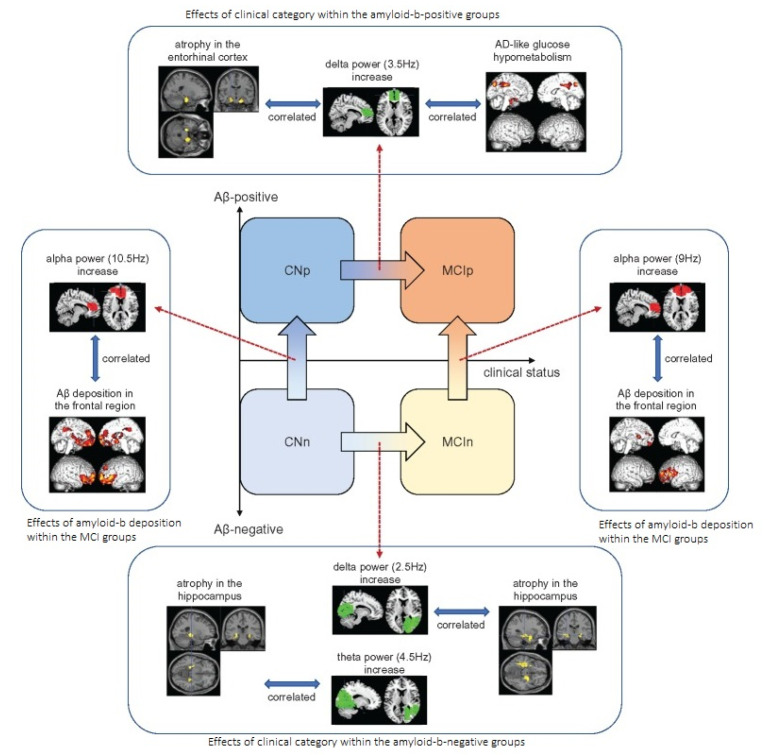
The characteristics of the MEG power markers. The arrows with the gradation colors indicate the directions where the relative power increases (not indicating clinical transition). Reprinted from [76]. (Copyright 2018 Oxford University Press) Image reproduced as per terms of CC BY-NC 4.0 license.

**Figure 7 brainsci-12-00788-f007:**
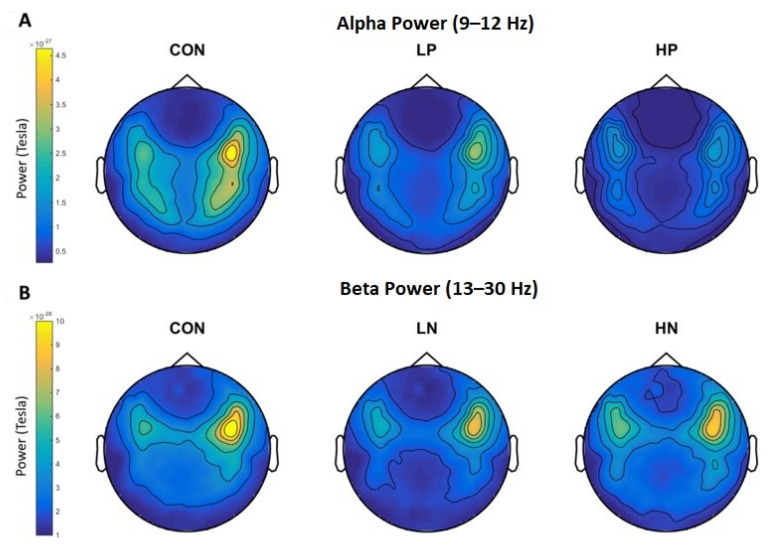
Topoplots of alpha power (**A**) for Con, Low positive (LP), and High Positive (HP); and of beta power (**B**) for Con, Low Negative (LN), and High Negative (HN) Schizophrenia patients. Colors represent power levels. Reprinted from [91]. (Copyright 2018 Elsevier) Image reproduced with copyright permission.

**Figure 8 brainsci-12-00788-f008:**
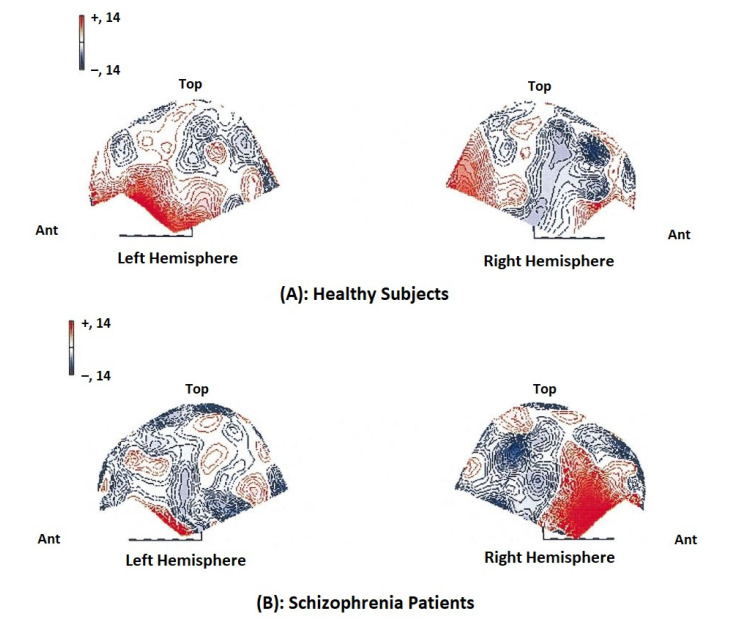
Difference map showing normalized gamma power (30 ± 45 Hz) in the mental arithmetic task minus normalized power at rest for controls (top row) and schizophrenia patients (bottom row). Power values are normalized according to McCarthy and Wood (1985). The red areas indicate an increase in power during cognitive activation. Reprinted from [92]. (Copyright 2000 Elsevier) Image reproduced with copyright permission.

**Figure 9 brainsci-12-00788-f009:**
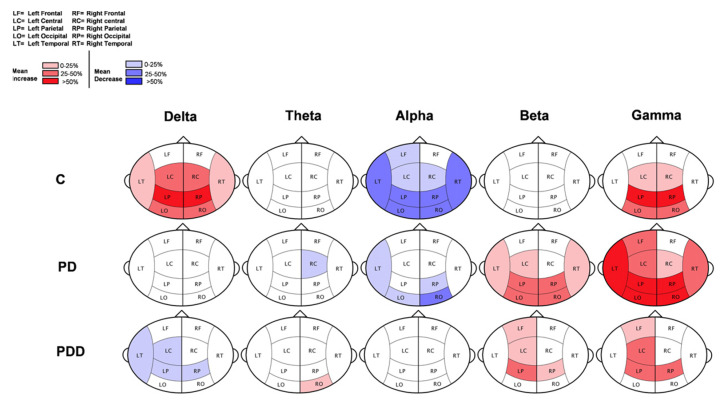
Schematic representation of results of the comparison between the eyes open and the eyes closed condition (PDD-—Parkinson’s disease-related dementia; PD–Parkinson’s disease without dementia; C—healthy, elderly controls. Reprinted from [107]. (Copyright 2006 Elsevier) Image reproduced with copyright permission.

**Table 1 brainsci-12-00788-t001:** Open source and licensed toolbox and their features for the analysis of MEG/EEG databases.

**Sl. No.**	**Toolbox**	**Features**	**Availability**
1	NUTMEG [43]	Supports MEG, EEG and intracranial EEG, Easy integration with other toolboxes GUI based functions Supports call function for batch analysis Tomographic visualisation is avaliable parametric and non-parametric statistics can be computed Functional Connectivity Mapping is available	open-source MATLAB-based toolbox
2	NIRS Brain AnalyzIR Toolbox [44]	Limited support for (EEG), (MEG), and surface-based fMRI (CIFTI) dense time-series data Contains modules like Pre-Processing, Data management filtering and first and second order statistical analysis GUI based functions	open-source MATLAB-based analysis
3	Field Trip [45]	Supports MEG, EEG, and Invasive Electrophysiological Data No GUI, hence, MATLAB command line scripting is possible Several types of time frequency analysis, connectivity analysis, and nonparametric statistical permutation tests at the channel and source level.	open-source MATLAB-toolbox
4	ELAN [46]	Supports MEG, EEG, and LFP Signals Analysis and Visualisation of data Supports Time Frequency analysis and Topographical Mapping. Capable of analysing Individual and group level statistics across the subjects Compatible with all types of MATLAB toolboxes like SPM, FieldTrip, Nutmeg, EEGLab and BrainStorm	Licenced version C implementation
5	SPM8 [47]	Supports MRI, fMRI, PET, MEG, EEG. Analysis and Visualisation of data Statistical Parametric Mapping of the analysed data Source Reconstruction Dynamic casual modelling for EEG and MEG	MATLAB-toolbox
6	Electro Magneto Encephalography Software [48]	Supports EEG and MEG data analysis and Visualisation Data pre-processing in EMEGS for statistical control of artifacts. Capable of analysing Statistical and Exploratory brain signals Supports ANOVA for region of interest analysis GUI based functions Synthetic data analysis for education.	MATLAB supported toolbox
7	Brainstorm [49]	Dedicated for EEG and MEG Signal Source estimation with MRI integration GUI based functions Supports MRI, EEG and MEG file formats Visualisation of Topological sensor data and anatomical structure volumes Registration and modelling of Multimodal data for analysis	Cross platform software supports MATLAB, Python and Java scripts
8	ERP WAVELAB [50]	Supports multichannel time frequency analysis of EEG and MEG data GUI based functions Supports scalp plotting with EEGLAB With ANOVA performs various statistical analysis.	open-source MATLAB-based analysis
9	MNE python toolbox [52].	Analysing and Visualisation of MEG, EEG, sEEG, ECoG, and NIRS data. Co-registration of MEG and MRI. Supports preprocessing, SSP, ICA, forward modeling, inverse methods, and Beamforming (Equivalent Current Dipole, Linearly Constrained Minimum-Variance) Supports time-frequency analysis, statistical analysis and connectivity estimation. GUI supported by MNELAB for MNE toolbox. Fast and memory efficient processing of large data sets	Open-source python package, Also avilable in MATLAB and C with limited modules.

**Table 2 brainsci-12-00788-t002:** MEG findings and Clinical Considerations.

Neurodisorders	MEG Findings and Clinical Considerations
Epilepsy	Accurate localization of spikes when compared with the EEG for both ictal and interictal subjects. It can localize the complex primary intrasylvian epileptiform disturbances associated with Landau–Kleffner syndrome, which aids the presurgical scenario [69]. MEG was found to be robust in the localization of postsurgical epileptiform disturbances [112]
Alzheimer’s Disease (AD)	Proficient in the early detection of dementia [113]. Increase in count of dipoles in the delta and theta band [114]. Slow wave activity detection in the right temporal and parietal lobe of the brain [115].
Schizophrenia	Resting-state activity was acquired spontaneously with 5 min duration in the awake state, resting state MEG are able to distinguish different subtypes of schizophrenia [116]. Auditory studies are also done with various stimuli for the distinguishing subtypes of schizophrenia [117]. MEG along with coherence source imaging (CSI) efficiently detects the brain oscillation that distinguishes between normal and schizophrenia subjects [2].
Parkinson Disease (PD)	Changes in beta band were observed in MEG data, PD patients had a significant minimization in beta ERD during the NoGo condition and in beta ERS during both Go and NoGo conditions compared with the healthy subjects [118]. Beta gamma phase magnitude coupling was observed in the resting state [119].

## Data Availability

Not applicable.

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
