# Peer review of "A Brief Introduction to Magnetoencephalography (MEG) and Its Clinical Applications"

_brainsci, 2022, doi:10.3390/brainsci12060788_

Round 1

Reviewer 1 Report

Overall nice work summarizing MEG applications and principles, though novelty is modest. It would be useful to cite work by Susan Bowyer on her work in autism and MEG as well as language lateralization and finally work by Abhimanyu Mahajan on MEG and dystonia/cervical dystonia.

Author Response

We would like to thank the reviewer for his insight and valuable feedback. We have incorporated the following paragraph as desired to incorporate the works. "The autism patients when subjected to eye-gaze processing have impaired activity in the gamma frequency band [23]. The authors concluded that the participants with severe ASD have higher activity between left temporo-parieto-occipital regions at 0-15 Hz frequency and higher activity between right temporo-parieto-occipital regions at 30 to 45 Hz in the low gamma frequency [24]. The 16 children with ASD are very sensitive to illegal speech sequences when undergoing 504.63Hz MEG recording. The participants are age- and gender-matched controls [25]. The ASD patients have low social behaviour and communication due to a pattern of lower gamma band coherence in angular and middle temporal cortical regions within the default mode network [26]. The study with DTI connectivity of hemisphere containing the epileptic focus in WM fibres of mTLE patients was confined with connectivity-based laterality model affecting these brain regions [27]. The authors studied the importance of Language mapping with MEG and the need for localization and lateralization with the changes in language networks and to identify the speech and social communication cortices in the brain [28]."

Reviewer 2 Report

  1. The title should mention the article type. ‘‘E.g. Narrative review’’
  2. Abstract. This structure ‘‘The outcome of the present work will aid researchers to utilize MEG in diagnostics.’’ Should be removed or rephrased. This is a narrative review without a systematic approach. Thus, it is advised not to provide direct conclusions.
  3. Grammatical analysis. There are some grammatical errors throughout the manuscript that need to be corrected. It is advised a cautious analysis or preferred editing service.
  4. ‘’over 10(14) synapses that assist in communication’’. It would be more understandable if the number description is provided.
  5. Table 1 is unnecessary. It is common knowledge based on the journal scope.
  6. The copyright permission for all the images should be provided as non-published supplementary material. This will avoid future issues related to copyright.
  7. A table with the clinical implications should be done. Some variables: Disorder, MEG findings, and clinical considerations.
  8. It is advised to provide two other specific chapters. One section should be about the disadvantages of such equipment. Also, future directions about the use of MEG should be done. How can MEG influence the development of future protocols or even guidelines?

Author Response

We would like to thank the reviewer for his insight and valuable feedback. Below are the changes we incorporated in our manuscript according to the feedback:

  1. We mentioned the article type as "Narrative review" as kindly directed by the reviewer.
  2. We had rephrased the sentence to " The outcome of this narrative review will aid researchers to utilize MEG in diagnostics."
  3. We earnestly apologize for the inadvertent grammatical errors. We have proof-read the manuscript in the revised copy and fixed all issues.
  4. We would like to thank the reviewer for this helpful pointer. Accordingly, we have made the following change in the manuscript. " The human brain is a complex organ, which consists of about 86 billion neurons and over (2.42 ± 0.29) ×10(14) synapses that assist in communication between the neurons[4]."
  5. Agreed. We have removed Table 1 as directed.
  6. We would like to apologize for having missed out in mentioning the copyright permission and we would thank the reviewer for pointing it out. Reproduction permissions granted have been mentioned in caption in the revised manuscript.
  7. We would like to thank the reviewer for his clear guideline. We have incorporated the following table as directed:

Table 2: MEG findings and Clinical Considerations

Neurodisorders

MEG findings and Clinical Considerations

Epilepsy

Accurate localization of spikes, when compared with the EEG for both ictal and interictal subjects. It can localize the complex primary intrasylvian epileptiform disturbances associated with Landau–Kleffner syndrome that aids the presurgical scenario [68]. MEG was found to be robust in the localization of postsurgical epileptiform disturbances [111]

Alzheimer’s Disease (AD) 

Proficient in the early detection of dementia [112]. Increase in count of dipoles in the delta and theta band [113]. Slow wave activity detection in the right temporal and parietal lobe of the brain [114].

Schizophrenia

Resting state activity was acquired spontaneously with 5 minutes duration in the awake state, resting state MEG are able to distinguish different subtypes of schizophrenia[115]. Auditory studies are also done with various stimuli for the distinguishing subtypes of schizophrenia[116]. MEG along with coherence source imaging (CSI) efficiently detects the brain oscillation that distinguishes between normal and Schizophrenia subjects [2].

Parkinson Disease (PD)

Changes in beta band were observed in MEG data, PD patients had a significant minimization in beta ERD during the NoGo condition and in beta ERS during both Go and NoGo conditions compared with the healthy subjects [117]. Beta gamma phase magnitude coupling was observed in the resting state[118].

            8. We have added the following chapter in the revised manuscript as per                  the kind suggestion of the reviewer.

"5. Inferences from the narrative study and Future scope of MEG

The following are the inferences from the research studies on MEG. The MEG signals of interest are extremely small, several orders of magnitude smaller than other signals in a typical environment that can obscure the signal. Thus, specialized shielding is required to eliminate the magnetic interference found in a typical urban clinical environment. Patients need to remain relatively still during a MEG exam. [109]. Some of the MEG studies rely on sensor space data and some on source space data, hence a generalized framework is required in the analysis[110]. There is a stimulus variation in most of the studies also and variation in output is observed prior to and after registration. MEG systems are less common when compared with the EEG and MRI systems, a standardization is also required in the quantitative analysis of MEG signals like protocols, data collection and data analysis. Few MEG public databases are available for research purposes, when compared with the EEG and MRI databases.

The usage of optically pumped magnetometers (OPMs) in future will generate proficient results, however, the movement of OPM relative to the scalp during acquisition and recording will generate artifacts. The MEG system based on OPM can generate results with higher spatial resolution than the brain since multichannel recording is possible with potential measurement near the brain. The wearable MEG in future will make the system simple, squid sensors are not required. The OPM based MEG will be beneficial in acquiring the signals from children within a duration of 10minutes, within the tolerance limits. Though MEG was proficient in the detection of neuro disorders, a specific pipeline for MEG data acquisition, processing and analysis in a clinical setting is required, since it is not possible to compare the results of MEG studies. The utilization of optical co-registration in future will also improve the accuracy in the localization of potential."

Reviewer 3 Report

This manuscript first briefly introduced the setup of Magnetoencephalography, and then comprehensively reviewed its clinical applications of the diagnosis of different neurological disorders, such as epilepsy, Alzheimer’s, Parkinsonism, autism, and schizophrenia, respectively. The overall structure of this manuscript is clear. However, some comments may need to be addressed before this manuscript is being considered for publication.

  • In Abstract, it was mentioned that MEG has higher SNR and spatial resolution than EEG. Can the authors use exact numbers? Can the authors also clarify how they forecasted the role of computer-aid algorithms in diagnosis and prediction of neurological disease?
  • In Introduction, it would be great if the author could discuss the advantages of MEG over other neuroimaging approaches, such as fMRI and PET.
  • The figures in this paper are all very blurry. Can the author use higher resolution figures?
  • Line 114, what the EOG stands for?
  • In 3.1 Signal Space Separation, it was pointed out that gradiometric coil configurations, Signal space projection (SSP), reference sensors and Signal Space Separation (SSS) are typical methods for removing noises. Can the author clarify why only SSS was discussed? Was it because it is the best technique than others?
  • Line 257-259 and line 311-313, “This section may be divided by subheadings. It should provide a concise and precise description of the experimental results, their interpretation, as well as the experimental conclusions that can be drawn.”. Duplicated and confusing sentence and it doesn’t seem to make sense to have it here.
  • Do the authors have reproduction permission to use Figures. 1-6? If so, please mention it in figure captions.
  • Figure 9 caption: it said that the figure was from [92], which is “A combined study of MEG and pico-Tesla TMS on children 727 with autism disorder”. However, the caption described Parkinson’s disease. Please clarify.
  • It looks like the Figure 5 is for epilepsy. Why was it referred in line 374 in the Alzheimer section?
  • Line 536, is epilepsy a type of neurodegenerative diseases?

Author Response

We would like to thank the reviewer for his insight and valuable feedback. We have incorporated the following changes as directed by the feedback. 

  • In Abstract, it was mentioned that MEG has higher SNR and spatial resolution than EEG. Can the authors use exact numbers? We have added the following snippet in the updated manuscript. The proven literature research works are there stating the importance of the computer aided algorithms in diagnosis and prediction of neurological disease. " The signal-to-noise ratio (SNRMEG =2.2 db, SNREEG <1 db) and spatial resolution (SRMEG =2-3 mm, SREEG =7-10 mm) is higher for MEG than EEG, hence MEG potentially facilitates accurate monitoring of cortical activity."
  • In Introduction, it would be great if the author could discuss the advantages of MEG over other neuroimaging approaches, such as fMRI and PET. We would like to thank the reviewer for his insight. Accordingly, we have made the following additions in the updated manuscript. "The temporal resolution of MEG was found to be better than fMRI and PET, since their resolution is on the order of seconds. MEG picks the fields generated by intraneuronal currents and hence gives a direct index of neuronal activity and synaptic current [3]"
  • The figures in this paper are all very blurry. Can the author use higher resolution figures? We have used higher resolution figures as per kind suggestions of the reviewer.
  • Line 114, what the EOG stands for? EOG stands for Electrooculography. We have added the expansion in the text at its first occurrence.
  • In 3.1 Signal Space Separation, it was pointed out that gradiometric coil configurations, Signal space projection (SSP), reference sensors and Signal Space Separation (SSS) are typical methods for removing noises. Can the author clarify why only SSS was discussed? Was it because it is the best technique than others? We would like to thank the reviewer for his insight. Accordingly, we have made the following additions in the updated manuscript. "SSS was found to be more proficient in improving the quality of MEG data than classical methods with less user intervention. This method proves to be robust and provides a shielding factor of about 150 and 50 for sources at 1 and 0.5m respectively [37]."
  • Line 257-259 and line 311-313, “This section may be divided by subheadings. It should provide a concise and precise description of the experimental results, their interpretation, as well as the experimental conclusions that can be drawn.”. Duplicated and confusing sentence and it doesn’t seem to make sense to have it here. We would like to apologize for the erroneous inclusion of that sentence. We have purged that statement in the revised manuscript.
  • Do the authors have reproduction permission to use Figures. 1-6? If so, please mention it in figure captions. Yes. Reproduction permissions have been granted and we have mentioned that in the relevant caption in the revised manuscript.
  • Figure 9 caption: it said that the figure was from [92], which is “A combined study of MEG and pico-Tesla TMS on children 727 with autism disorder”. However, the caption described Parkinson’s disease. Please clarify. We would like to apologize for the typo. We have amend the reference in the revised manuscript as follows. "106. Bosboom JLW, Stoffers D, Stam CJ, et al (2006) Resting state oscillatory brain dynamics in Parkinson’s disease: an MEG study. Clinical Neurophysiology 117:2521–2531"
  • It looks like the Figure 5 is for epilepsy. Why was it referred in line 374 in the Alzheimer section? We would like to apologize for the typo. We have fixed the issue in the revised manuscript.
  • Line 536, is epilepsy a type of neurodegenerative diseases? We would like to apologize for the typo. We have fixed the issue in the revised manuscript. "This review gives a basic introduction about MEG and its clinical applications for neurodegenerative and associated diseases like epilepsy, Alzheimer, schizophrenia, and Parkinson’s diseases."

Round 2

Reviewer 2 Report

The present narrative review about MEG is interesting. The authors greatly improved the quality of the manuscript by providing copyrights, including table 2, and discussing future perspectives. The references need to be adjusted according to the page.

Reviewer 3 Report

The authors have solved my concerns. Recommend for publication.